# Trends of Antimicrobial Susceptibility in Clinically Significant Coagulase-Negative Staphylococci Isolated from Cerebrospinal Fluid Cultures in Neurosurgical Adults: a Nine-Year Analysis

Yi Ye,[a] Ye Tian,[a] Yueyue Kong,[a] Jiawei Ma,[a] Guangzhi Shi[a]

[a]Department of Critical Care Medicine, Beijing Tiantan Hospital, Capital Medical University, Beijing, China

Yi Ye and Ye Tian contributed equally to this article. Author order was determined in order of decreasing seniority.

**ABSTRACT** Coagulase-negative staphylococci (CoNS) are the main pathogens in health care-associated ventriculitis and meningitis (HCAVM). This study aimed to assess antimicrobial susceptibility. Moreover, the treatment and clinical outcome were described. All neurosurgical adults admitted to one of the largest neurosurgical centers in China with clinically significant CoNS isolated from cerebrospinal fluid cultures in 2012 to 2020 were recruited. One episode was defined as one patient with one bacterial strain. Interpretive categories were applied according to the MICs. The clinical outcomes were dichotomized into poor (Glasgow Outcome Scale 1 to 3) and acceptable (Glasgow Outcome Scale 4 to 5). In total, 534 episodes involving 519 patients and 16 bacteria were analyzed. Over the 9 years, eight antimicrobial agents were used in antimicrobial susceptibility tests, including six in over 80% of CoNS. The range of resistance rates was 0.8% to 84.6%. The vancomycin resistance rate was the lowest, whereas the penicillin resistance rate was the highest. The linezolid (a vancomycin replacement) resistance rate was 3.1%. The rate of oxacillin resistance, representing methicillin-resistant staphylococci, was 70.2%. There were no significant trends of antimicrobial susceptibility over the 9 years for any agents analyzed. However, there were some apparent changes. Notably, vancomycin-resistant CoNS appeared in recent years, while linezolid-resistant CoNS appeared early and disappeared in recent years. Vancomycin (or norvancomycin), the most common treatment agent, was used in 528 (98.9%) episodes. Finally, 527 (98.7%) episodes had acceptable outcomes. It will be safe to use vancomycin to treat CoNS-related HCAVM in the immediate future, although continuous monitoring will be needed.

**IMPORTANCE** Coagulase-negative staphylococci are the main pathogens in health care-associated ventriculitis and meningitis. There are three conclusions from the results of this study. First, according to antimicrobial susceptibility, the rates of resistance to primary antimicrobial agents are high and those to high-level agents, including vancomycin, are low. Second, the trends of resistance rates are acceptable, especially for high-level agents, although long-term and continuous monitoring is necessary. Finally, the clinical outcomes of neurosurgical adults with coagulase-negative staphylococci-related health care-associated ventriculitis and meningitis are acceptable after treatment with vancomycin. Therefore, according to the antimicrobial susceptibility and clinical practice, vancomycin will be safe to treat coagulase-negative staphylococci-related health care-associated ventriculitis and meningitis.

**KEYWORDS** antimicrobial susceptibility test, coagulase-negative staphylococci, MIC, resistance

Coagulase-negative staphylococci (CoNS) are the main pathogens in health care-associated ventriculitis and meningitis (HCAVM) (1–5).

The problem of antimicrobial-resistant CoNS isolated from cultures from patients is severe. Several studies have reported that over 90% of CoNS are resistant to penicillin,

Address correspondence to Guangzhi Shi, shiguangzhi@bjtth.org.

The authors declare no conflict of interest.

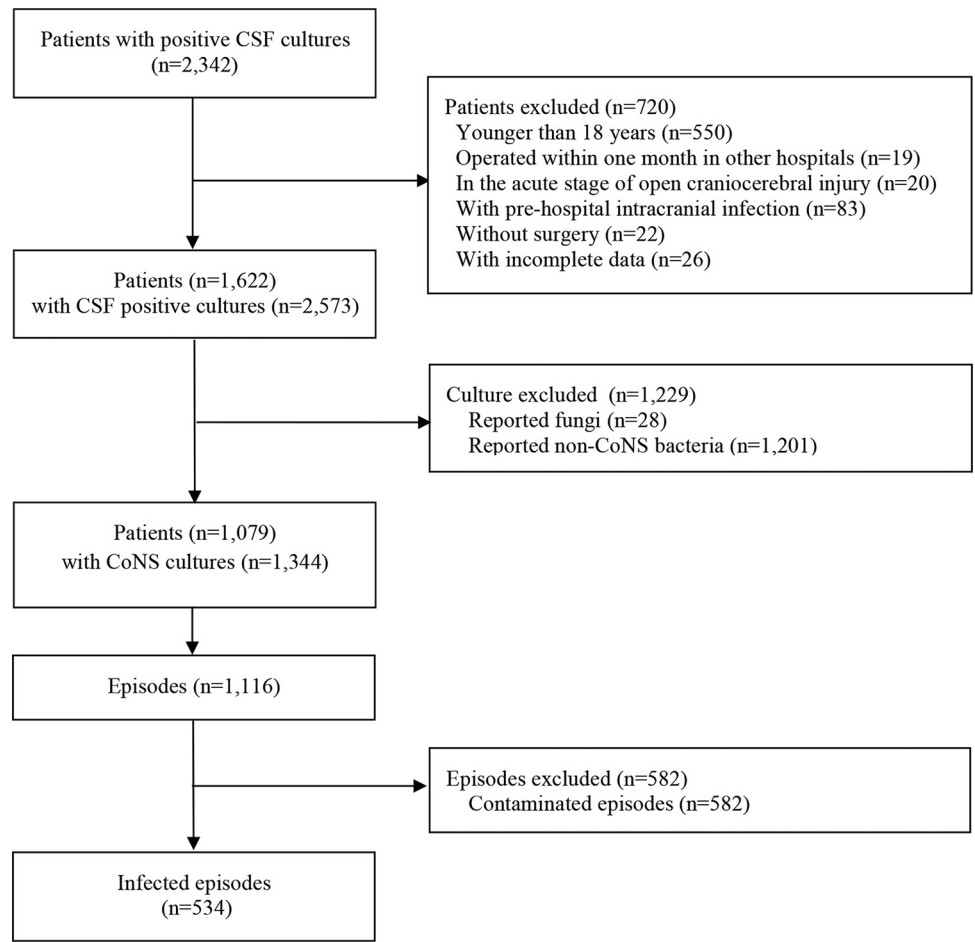

**FIG 1** Flow chart. CSF: cerebrospinal fluid; CoNS: coagulase-negative staphylococci; episode: one episode was defined as one patient with one strain of bacteria.

64.2% to 93.6% of CoNS are resistant to methicillin, and all CoNS are susceptible to vancomycin (6–9). In other studies, vancomycin- and linezolid-resistant CoNS have occasionally been reported (10–12). The trends are not favorable. Several studies have reported that the resistance rates have significantly increased, especially for oxacillin (13–16).

However, large-sample collections of clinically significant CoNS isolated from cerebrospinal fluid (CSF) cultures in neurosurgical patients have not been specifically analyzed. The results of previous studies are not representative of the CoNS causing HCAVM for two reasons. First, over half of CoNS isolated from CSF cultures are not clinically significant (17–19). It is not suitable to represent clinically significant CoNS with all CoNS. Second, the duration of hospital stay of neurosurgical patients is long, and colonization by antimicrobial-resistant bacteria occurs frequently (20, 21), followed by infection. Therefore, the CoNS causing HCAVM could have higher resistance rates than general CoNS.

To comprehensively understand the clinically significant CoNS isolated from CSF cultures in neurosurgical patients, we analyzed the results and trends of antimicrobial susceptibility. Moreover, the treatment and clinical outcome were described.

## RESULTS

**Patients.** A total of 2,342 patients were recruited. After exclusion, 534 episodes from 519 patients were analyzed (Fig. 1).

The mean age was 44.6 ± 13.3 years (range from 18 to 80 years) for all episodes, 226 (42.3%) episodes involved females, and patients composing 443 (83.0%) episodes were diagnosed with solid tumors. Three (0.6%) episodes had ventriculitis, and the remaining episodes had meningitis (Table 1).

**TABLE 1** Demographic characteristics of episodes with different bacteria

| Characteristic | Total (n = 534) | S. epidermidis (n = 273) | S. hominis (n = 90) | S. haemolyticus (n = 62) | S. capitis (n = 47) | Other[a] (n = 62) | P value |
|---|---|---|---|---|---|---|---|
| Age (yr), mean ± SD | 44.6 ± 13.3 | 44.5 ± 13.4 | 46.9 ± 12.4 | 44.4 ± 12.8 | 43.0 ± 13.6 | 43.1 ± 14.0 | 0.328 |
| Female, n (%) | 226 (42.3) | 129 (47.3) | 39 (43.3) | 16 (25.8) | 18 (38.3) | 24 (38.7) | 0.035 |
| Main diagnosis, n (%) | | | | | | | |
| Solid tumor | 443 (83.0) | 224 (82.1) | 79 (87.8) | 48 (77.4) | 41 (87.2) | 51 (82.3) | 0.461 |
| Vascular malformation | 52 (9.7) | 30 (11.0) | 6 (6.7) | 6 (9.7) | 3 (6.4) | 7 (11.3) | 0.695 |
| Traumatic brain injury | 15 (2.8) | 7 (2.6) | 1 (1.1) | 4 (6.5) | 2 (4.3) | 1 (1.6) | 0.319 |
| Functional disease | 10 (1.9) | 3 (1.1) | 4 (4.4) | 2 (3.2) | 0 (0.0) | 1 (1.6) | 0.226 |
| Other diseases | 14 (2.6) | 9 (3.3) | 0 (0.0) | 2 (3.2) | 1 (2.1) | 2 (3.2) | 0.536 |
| Surgery, n (%) | | | | | | | |
| Craniotomy | 498 (93.3) | 250 (91.6) | 84 (93.3) | 60 (96.8) | 47 (100.0) | 57 (91.9) | 0.198 |
| Transsphenoidal surgery | 27 (5.1) | 16 (5.9) | 5 (5.6) | 2 (3.2) | 0 (0.0) | 4 (6.5) | 0.463 |
| Spinal surgery | 9 (1.7) | 7 (2.6) | 1 (1.1) | 0 (0.0) | 0 (0.0) | 1 (1.6) | 0.506 |
| Chronic diseases, n (%)[b] | 135 (25.3) | 69 (25.3) | 28 (31.1) | 13 (21.0) | 13 (27.7) | 12 (19.4) | 0.474 |
| Admission GCS[c], n (%) | | | | | | | |
| 13–15 | 523 (97.9) | 268 (98.2) | 88 (97.8) | 58 (93.5) | 47 (100.0) | 62 (100.0) | 0.081 |
| 9–12 | 4 (0.7) | 1 (0.4) | 0 (0.0) | 3 (4.8) | 0 (0.0) | 0 (0.0) | 0.003 |
| 3–8 | 7 (1.3) | 4 (1.5) | 2 (2.2) | 1 (1.6) | 0 (0.0) | 0 (0.0) | 0.714 |
| CSF[d] leak, n (%) | 53 (9.9) | 32 (11.7) | 8 (8.9) | 6 (9.7) | 2 (4.3) | 5 (8.1) | 0.553 |
| Incision infection, n (%) | 39 (7.3) | 22 (8.1) | 6 (6.7) | 4 (6.5) | 3 (6.4) | 4 (6.5) | 0.976 |
| ICU, n (%)[e] | 50 (9.4) | 28 (10.3) | 6 (6.7) | 4 (6.5) | 1 (2.1) | 11 (17.7) | 0.046 |
| With other bacteria, n (%)[f] | 71 (13.3) | 31 (11.4) | 13 (14.4) | 9 (14.5) | 4 (8.5) | 14 (22.6) | 0.156 |
| Severe infection, n (%)[g] | 7 (1.3) | 2 (0.7) | 4 (4.4) | 1 (1.6) | 0 (0.0) | 0 (0.0) | 0.060 |
| Resistance, n (%)[h] | 4 (0.7) | 3 (1.1) | 0 (0.0) | 0 (0.0) | 1 (2.1) | 0 (0.0) | 0.514 |
| Only vancomycin, n (%)[i] | 506 (94.8) | 258 (94.5) | 87 (96.7) | 60 (96.8) | 46 (97.9) | 55 (88.7) | 0.154 |
| Poor outcome, n (%) | 7 (1.3) | 4 (1.5) | 2 (2.2) | 0 (0.0) | 0 (0.0) | 1 (1.6) | 0.714 |

[a]Including *Staphylococcus warneri* (n = 16), *Staphylococcus lugdunensis* (n = 9), unclassified coagulase-negative staphylococci (n = 8), *Staphylococcus saprophyticus* (n = 7), *Staphylococcus cohnii* (n = 7), *Staphylococcus caprae* (n = 4), *Staphylococcus pettenkoferi* (n = 3), *Staphylococcus gallinarum* (n = 2), *Staphylococcus equorum* (n = 2), *Staphylococcus simulans* (n = 2), *Staphylococcus xylosus* (n = 1), and *Staphylococcus sciuri* (n = 1).
[b]Including hypertension in 100 episodes, diabetes mellitus in 20 episodes, coronary heart disease in 8 episodes, hepatitis B in 6 episodes, and rare diseases, including hyperthyroidism, hypothyroidism, hyperlipidemia, etc.
[c]Glasgow coma scale.
[d]Cerebrospinal fluid.
[e]Means in intensive care unit more than 24 h.
[f]With other bacteria isolated from CSF cultures.
[g]Including four abscesses and three ventriculitis.
[h]Means resistant to vancomycin.
[i]Only accept vancomycin as treatment for coagulase-negative staphylococci-related health care-associated ventriculitis and meningitis.

**Bacteria.** There were 16 bacteria, including 15 bacterial species and unclassified CoNS. *Staphylococcus epidermidis*, the most common bacterial species, accounted for 273 (51.1%) episodes. The number of clinically significant CoNS in 2019 was the largest because the hospital moved to a new campus with more beds in October 2018. Then, the number in 2020 decreased because of the coronavirus disease 2019 pandemic (Fig. 2).

**Antimicrobial agents.** There were eight antimicrobial agents used in the antimicrobial susceptibility tests. The relationships of interpretive categories and MICs are shown in Table 2. Six agents were used in over 80% of CoNS, and two agents were used in less than 50% of CoNS (Table 3).

**Results of antimicrobial susceptibility tests.** Resistance rates were different for different agents. The range of resistance rates was 0.8% to 84.6%. The vancomycin resistance rate was the lowest, and the penicillin resistance rate was the highest. The linezolid (a vancomycin replacement) resistance rate was 3.1%. The rate of oxacillin resistance, representing methicillin-resistant staphylococci, was 70.2%, and the rifampin and teicoplanin resistance rates were 4.4% and 3.7%, respectively (Table 3).

**Trends of antimicrobial susceptibility.** There were no significant trends of antimicrobial susceptibility for any agents analyzed during the 9-year period. However, there were apparent changes in some years (Fig. 3).

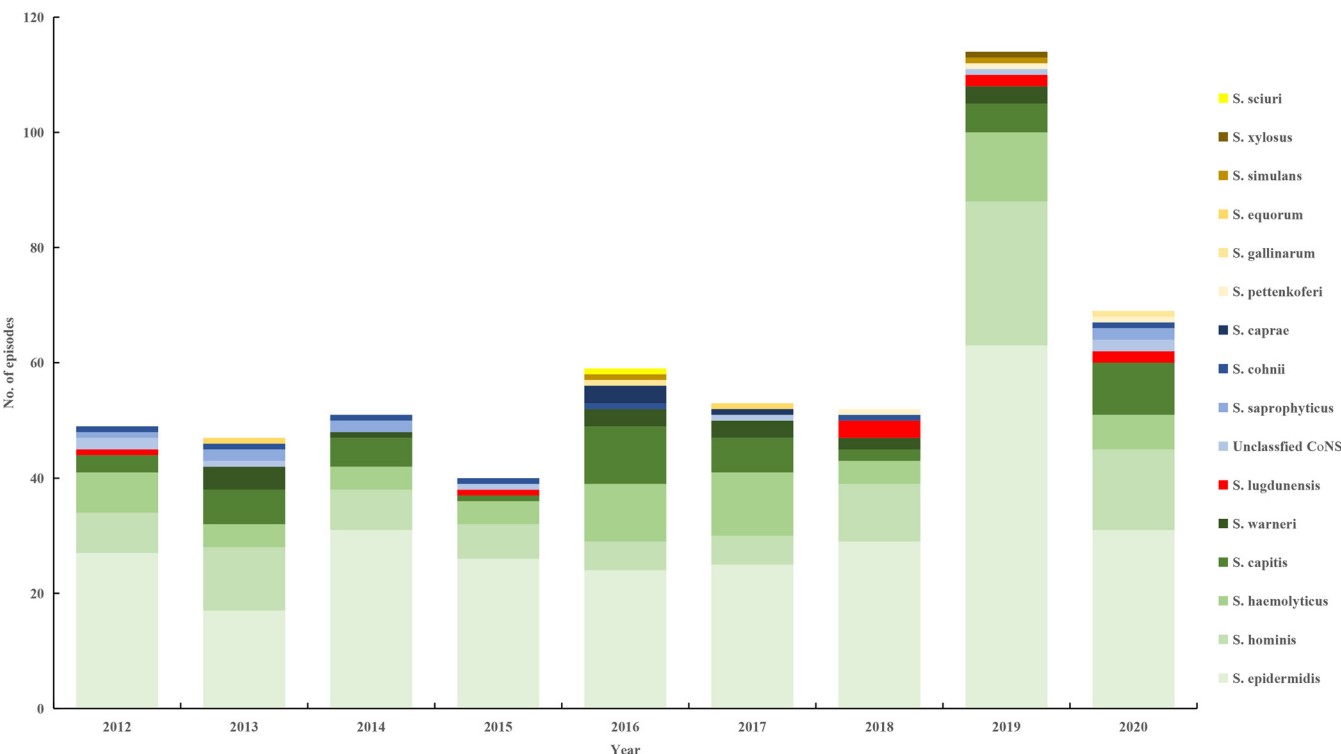

**FIG 2** No. of episodes with clinically significant bacteria in the 9 years. S, *Staphylococcus*; CoNS, coagulase-negative staphylococci.

Vancomycin-resistant CoNS appeared in recent years. In contrast, linezolid-resistant CoNS appeared early in the study and disappeared in recent years. For oxacillin, the resistance rate increased from 2016 to 2019 but decreased in 2020, especially for MICs of ≥4 μg/mL. For penicillin, the resistance rate decreased in recent years. For rifampin and teicoplanin, the resistance rates gradually decreased to zero. For gentamicin and trimethoprim-sulfamethoxazole, the resistance rates increased in recent years (Fig. 3).

**Treatment.** Vancomycin (or norvancomycin), the most common antimicrobial agent in clinical practice, was used in 528 (98.9%) episodes. However, the treatment was adjusted in 22 episodes because of poor or adverse effects. Linezolid was used in 21 episodes, including 4 episodes with vancomycin-resistant CoNS, and teicoplanin was used in 1 episode.

Only five episodes involved the use of linezolid as the whole-course treatment, and one episode used teicoplanin in such a manner.

**Clinical outcome.** A total of 527 (98.7%) episodes had acceptable outcomes. The seven episodes that had poor outcomes involved CoNS susceptible to vancomycin and linezolid.

**TABLE 2** The interpretive categories based on MICs[a]

| Agents | Susceptible | Intermediate[b] | Resistant |
|---|---|---|---|
| Oxacillin for other CoNS[c] | ≤0.5 | | ≥1 |
| Oxacillin for *Staphylococcus lugdunensis* | ≤2 | | ≥4 |
| Gentamicin | ≤4 | 8 | ≥16 |
| Penicillin | ≤0.125 | | ≥0.25 |
| Rifampin | ≤1 | 2 | ≥4 |
| Linezolid | ≤4 | | ≥8 |
| Vancomycin | ≤4 | 8–16 | ≥32 |
| Teicoplanin | ≤8 | 16 | ≥32 |
| Trimethoprim-sulfamethoxazole | ≤2/38 | | ≥4/76 |

[a]All data are MICs (μg/mL).
[b]Susceptible-dose dependent.
[c]Coagulase-negative staphylococci.

**TABLE 3** The agents used in antimicrobial susceptibility tests and *n* (%) of resistant isolates in the 9 years

| Agent | No. of resistant isolates (%) in: | | | | | | | | | |
|---|---|---|---|---|---|---|---|---|---|---|
| | Total | 2012 | 2013 | 2014 | 2015 | 2016 | 2017 | 2018 | 2019 | 2020 |
| Episodes | 534 | 49 | 47 | 51 | 40 | 59 | 53 | 52 | 114 | 69 |
| Oxacillin | 524 | 48 | 45 | 47 | 40 | 58 | 53 | 52 | 114 | 67 |
| R[a] | 368 (70.2) | 35 (72.9) | 34 (75.6) | 35 (74.5) | 23 (57.5) | 33 (56.9) | 40 (75.5) | 40 (76.9) | 88 (77.2) | 40 (59.7) |
| Gentamicin | 523 | 48 | 44 | 48 | 40 | 58 | 53 | 52 | 113 | 67 |
| R | 90 (17.2) | 11 (22.9) | 9 (20.5) | 7 (14.6) | 8 (20.0) | 17 (29.3) | 15 (28.3) | 4 (7.7) | 10 (8.8) | 9 (13.4) |
| Penicillin | 518 | 46 | 43 | 47 | 39 | 58 | 53 | 52 | 113 | 67 |
| R | 438 (84.6) | 39 (84.8) | 38 (88.4) | 39 (83.0) | 29 (74.4) | 46 (79.3) | 49 (92.5) | 47 (90.4) | 98 (86.7) | 53 (79.1) |
| Rifampin | 521 | 48 | 45 | 46 | 40 | 58 | 53 | 52 | 113 | 66 |
| R | 23 (4.4) | 3 (6.3) | 3 (6.7) | 1 (2.2) | 4 (10.0) | 6 (10.3) | 2 (3.8) | 2 (3.8) | 2 (1.8) | 0 (0.0) |
| Linezolid | 520 | 48 | 45 | 47 | 40 | 58 | 53 | 52 | 111 | 66 |
| R | 16 (3.1) | 4 (8.3) | 3 (6.7) | 4 (8.5) | 2 (5.0) | 1 (1.7) | 2 (3.8) | 0 (0.0) | 0 (0.0) | 0 (0.0) |
| Vancomycin | 518 | 48 | 44 | 47 | 40 | 58 | 51 | 52 | 112 | 66 |
| R | 4 (0.8) | 0 (0.0) | 0 (0.0) | 0 (0.0) | 0 (0.0) | 0 (0.0) | 2 (3.9) | 0 (0.0) | 2 (1.8) | 0 (0.0) |
| Teicoplanin | 242 | 43 | 44 | 35 | 32 | 48 | 31 | 4 | 1 | 4 |
| R | 9 (3.7) | 4 (9.3) | 2 (4.5) | 0 (0.0) | 0 (0.0) | 1 (2.1) | 2 (6.5) | 0 (0.0) | 0 (0.0) | 0 (0.0) |
| SXT[b] | 248 | 48 | 45 | 37 | 33 | 47 | 34 | 4 | | |
| R | 115 (46.4) | 30 (62.5) | 19 (42.2) | 21 (56.8) | 16 (48.5) | 15 (31.9) | 12 (35.3) | 2 (50.0) | | |

[a]Resistant.
[b]Trimethoprim-sulfamethoxazole.

Four bacterial species were specifically analyzed in the cohort analysis, and the remaining species were analyzed as a group. The unadjusted analysis showed that the range of poor outcome rates was 0.0% to 2.2% (*P* = 0.714) (Table 1). The adjusted analysis based on *S. epidermidis* showed no association between the bacterial species and clinical outcome (Table 4).

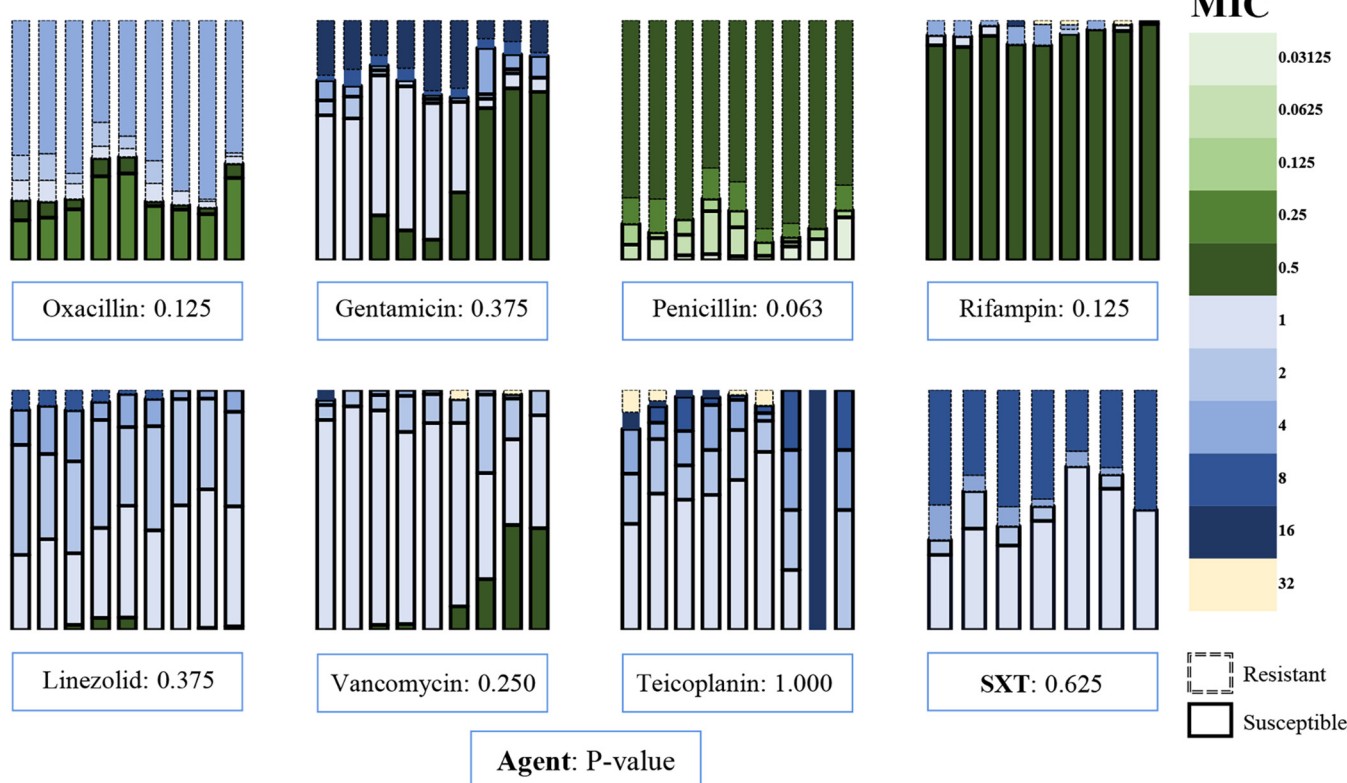

**FIG 3** The MICs (μg/mL) of antimicrobial agents in the 9 years. *P* value, for the trend of resistance rate; SXT, trimethoprim-sulfamethoxazole, the period was 2012 to 2018, and the MIC was based on trimethoprim.

**TABLE 4** Relative risk for poor outcome rate of episodes with different bacteria

| Bacteria | No. of episodes | Relative risk | 95% confidence interval | P value[a] |
|---|---|---|---|---|
| S. epidermidis | 273 | | | |
| S. hominis | 90 | 0.78 | 0.31–1.94 | 0.590 |
| S. haemolyticus[b] | 62 | | | 0.995 |
| S. capitis[b] | 47 | | | 0.996 |
| Other[c] | 62 | 1.86 | 0.37–9.45 | 0.452 |

[a]Compared with S. epidermidis.
[b]The poor outcome rate of episodes with S. haemolyticus or Staphylococcus capitis was 0.0%.
[c]Including Staphylococcus warneri (n = 16), Staphylococcus lugdunensis (n = 9), unclassified coagulase-negative staphylococci (n = 8), Staphylococcus saprophyticus (n = 7), Staphylococcus cohnii (n = 7), Staphylococcus caprae (n = 4), Staphylococcus pettenkoferi (n = 3), Staphylococcus gallinarum (n = 2), Staphylococcus equorum (n = 2), Staphylococcus simulans (n = 2), Staphylococcus xylosus (n = 1), and Staphylococcus sciuri (n = 1).

## DISCUSSION

Overall, the antimicrobial resistance of clinically significant CoNS isolated from CSF cultures in neurosurgical adults was severe, especially to primary antimicrobial agents, including penicillin and oxacillin. Thus, these agents are not suitable for treating CoNS-related HCAVM. Vancomycin, the leading empirical agent (22), is safe to treat CoNS-related HCAVM according to antimicrobial susceptibility. Linezolid, the replacement for vancomycin, is also clinically valuable. Moreover, rifampin and teicoplanin could be valuable agents. The results of the antimicrobial susceptibility tests are similar to those in previous studies (7–9, 11, 23, 24).

There was no significant trend for any agent over the 9-year study period, although the data showed trends for some years. Therefore, short-term trends do not predict long-term changes; thus, long-term and continuous monitoring is necessary.

According to the trends, although vancomycin-resistant CoNS were rare, continuous monitoring is needed, and perhaps more vancomycin-resistant CoNS will appear in the future, while linezolid will continue to be safe to treat CoNS-related HCAVM. The possible reason could be that linezolid was rarely used. The trends of rifampin and teicoplanin resistance rates were similar. This conclusion is different from those of previous studies that reported that resistance rates usually increase (13–16).

Vancomycin (or norvancomycin), linezolid, and teicoplanin were used to treat CoNS-related HCAVM. The most common agent was vancomycin (or norvancomycin). Linezolid was the usual replacement, and teicoplanin was used occasionally. The clinical outcome of patients with CoNS-related HCAVM was generally acceptable. Therefore, vancomycin (or norvancomycin) is safe to treat CoNS-related HCAVM in clinical practice, consistent with its performance in the antimicrobial susceptibility tests.

Moreover, the clinical outcome was not related to bacterial species. The relationship between resistance and clinical outcome cannot be discussed in this study for two reasons. First, the rates of resistance to antimicrobial agents in clinical practice were low, and the sample size was not large enough. Second, although the rates of resistance to primary agents were high, it is obvious that resistance cannot affect the clinical outcome unless resistance to primary agents is related to resistance to high-level agents.

There are some limitations to this study. First, this study is retrospective. Second, all agents were used for some portion of the CoNS, which may cause bias. Finally, the results were separated by year, and there may be additional ways to compare these results.

In total, we reached three conclusions in the first study about CoNS causing HCAVM in neurosurgical adults. First, the rates of resistance to primary antimicrobial agents are high, and those to high-level agents used in clinical practice are low. Second, the trends of resistance rates are acceptable, especially for high-level agents, although long-term and continuous monitoring will be needed. Finally, the clinical outcome of neurosurgical adults with CoNS-related HCAVM is acceptable, and vancomycin is safe to treat CoNS-related HCAVM according to antimicrobial susceptibility and clinical practice.

**Conclusion.** It will be safe to use vancomycin to treat CoNS-related HCAVM in the immediate future, although continuous monitoring will be needed.

## MATERIALS AND METHODS

**Study design and setting.** This retrospective study recruited all patients with positive CSF cultures from January 2012 to December 2020 in the Department of Neurosurgery in Beijing Tiantan Hospital, Capital Medical University (Beijing, China), a tertiary teaching hospital with one of the largest neurosurgical centers in China. All data were collected from electronic medical records.

Exclusion criteria for the patients included younger than 18 years, operated within 1 month in other hospitals, in the acute stage of open craniocerebral injury, with prehospital intracranial infection, without surgery, and with incomplete data.

**Cultures and antimicrobial susceptibility tests.** CSF specimens were collected from patients with suspected HCAVM. At the bedside, CSF was obtained by well-trained neurosurgical doctors after disinfection with 70% isopropyl alcohol followed by 10% povidone-iodine as antiseptics. The most common sampling method was lumbar puncture, followed by ventricular drainage and lumbar cistern drainage. All CSF specimens were immediately transferred to the laboratory and incubated until flagged as positive or for 5 days in a Bactec 9240 (Becton, Dickinson, America) for samples from January 2012 to September 2018 or a BacT/Alert 3D (bioMérieux, France) for samples from October 2018 to December 2020. The broth was analyzed automatically every 10 min; bacteria from positive samples were Gram stained and subcultured on solid medium using standard protocols.

Antimicrobial agents were tested for activity against the bacteria using disk diffusion and broth microdilution methods according to the Clinical and Laboratory Standards Institute. The tests were performed according to the newest editions at the corresponding times, and the interpretive categories were applied according to the Clinical and Laboratory Standards Institute M100, 31st ed. (25).

Exclusion criteria for the positive cultures were reported fungi and non-CoNS bacteria. Agents administered by the oral route only, 1st- and 2nd-generation cephalosporins, cephamycins, clindamycin, macrolides, tetracyclines, and fluoroquinolones were not analyzed because antimicrobial susceptibility tests were used to guide the treatment of HCAVM (25).

**Exclusion of not clinically significant CoNS.** An episode was defined as one patient with one strain of bacteria. Therefore, a patient with two or more strains of bacteria constituted two or more episodes.

An infected episode was defined as a patient who had clinical features of HCAVM (22) and an improvement related to a susceptible antimicrobial agent. However, the therapeutic effect of the susceptible antimicrobial agent alone could be deficient in patients with special situations, including poor wound healing (26), implants (27), CSF leakage (28), abscess (29), and ventriculitis (30). For episodes with two or more positive cultures, the first positive culture was analyzed (25).

The contaminated episode was defined according to the diagnostic criteria (Appendix 1). All CoNS belonging to contaminated episodes were not clinically significant and were excluded.

**Clinical outcome.** The clinical outcomes were dichotomized into poor (Glasgow Outcome Scale 1 to 3) and acceptable (Glasgow Outcome Scale 4 to 5). The judgment was performed during the hospitalization in which the patient had HCAVM. Unplanned readmission within 1 month was recognized as a continuation of the previous hospitalization.

The relationship of bacterial species and clinical outcome in a cohort analysis was discussed. The exposure was bacterial species, and the study outcome was a poor outcome rate. A bacterial species was discussed specifically when it existed in over 30 episodes. The episodes with the most common bacterial species were set as the nonexposed group in the adjusted analysis.

Apart from the exposure and the study outcome, demographic characteristics, basic health information, surgery history, intensive care unit admission, and infection-related information were collected and analyzed.

**Statistical analysis.** Categorical variables are presented as frequencies and percentages. Continuous variables were described using means and standard deviations. Statistical analyses were performed using R Programming Language version 4.0.2. Characteristics of episodes were compared using contingency analysis for categorical variables and the Kruskal–Wallis rank-sum test for continuous variables. The Cox-Stuart test was used to detect trends in the resistance rate. The adjusted analysis was performed using a generalized linear model. $P$ values of $<0.05$ were significant.

## SUPPLEMENTAL MATERIAL

Supplemental material is available online only.

**SUPPLEMENTAL FILE 1**, PDF file, 0.1 MB.

## ACKNOWLEDGMENT

This work was supported by the Yangfan Plan of Beijing Municipal Hospital Administration (ZYLX202109).

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
