## [Reviewer comments · Microbiology Spectrum]

Microbiology Spectrum

Trends of antimicrobial susceptibility tests in clinically significant coagulase-negative staphylococci isolated from cerebrospinal fluid cultures: a nine-year analysis

Yi Ye, Ye Tian, Yueyue Kong, Jiawei Ma, and Guangzhi Shi

Corresponding Author(s): Guangzhi Shi, Beijing Tiantan Hospital, Capital Medical University

Review Timeline:

Submission Date:	September 4, 2021
Editorial Decision:	November 12, 2021
Revision Received:	December 2, 2021
Editorial Decision:	December 15, 2021
Revision Received:	December 28, 2021
Accepted:	January 4, 2022

Editor: Tomefa Asempa

Reviewer(s): Disclosure of reviewer identity is with reference to reviewer comments included in decision letter(s). The following individuals involved in review of your submission have agreed to reveal their identity: Vladimir Gostev (Reviewer #1)

Transaction Report:

DOI: <https://doi.org/10.1128/spectrum.01462-21>

November 12, 2021

Prof. Guangzhi Shi
Beijing Tiantan Hospital, Capital Medical University
No.119, South Fourth Ring West Road, Fengtai District, Beijing
Beijing
China

Re: Spectrum01462-21 (Trends of antimicrobial susceptibility tests in clinically significant coagulase-negative staphylococci isolated from cerebrospinal fluid cultures: a nine-year analysis)

Dear Prof. Guangzhi Shi:

Link Not Available

Sincerely,

Tomefa Asempa

Journals Department
Reviewer comments:

Reviewer #1 (Comments for the Author):

In the study of M.M. Yi Ye et al. the retrospective analysis of dynamic antibiotic susceptibility of clinical isolates of coagulase-negative staphylococci (CoNS) isolated from the patients with neurological infections was performed. In order to differentiate the contaminated samples and true infections inclusion criteria were applied. Since the work is exclusively retrospective and the authors used the results of lab information systems only, no experimental studies were carried out. I've got the feeling that the manuscript is more like a report than a research study. Perhaps the work should be supplemented with some clinical data - a detailed description of clinical outcomes, its associations with different species of CoNS, as well as some additional clinical parameters for a proper comparison. Which treatment have the patients received? Was there any correlation with the antibiotic susceptibility data? What was the time period during the treatment after which the cerebrospinal fluid was free of CoNS?

Minor comments:

English should be revised.
Which method was used for CoNS identification?
Versions of CLSI, EUCAST should be specified.

Reviewer #2 (Comments for the Author):

This work by Ye et al. aims to investigate the trends of antimicrobial susceptibility in clinically significant coagulase-negative staphylococci (CoNS) isolated from cerebrospinal fluid (CSF) cultures in a Chinese hospital over a period of 9 years. The authors have concluded that no significant trend of antimicrobial susceptibility was observed in CoNS against the antimicrobials analysed in the study over the period of 9 years. However, the authors have also noted some visible changes, such as the increasing of vancomycin resistance and the 'disappearing' of lizezolid resistance. Although the study informs the overall situation of antimicrobial resistance in clinical CoNS in the Chinese hospital, much is needed to improve the methodology and data presentation in this manuscript in order to support these conclusions.

Major comments:

1. Most of the recruited 2342 patients were excluded (leaving one 519 patients). What is the rationale behind the patient exclusion criteria? For instance, if the authors were to investigate the antimicrobial susceptibility in clinically significant CoNS, why cultures from patients under 18 were excluded?
2. What is a contaminated episode? Figure 1 shows 1344 cultures were positive with CoNS amongst 1116 episodes, if I understand correctly. Then it shows 582 episodes were excluded because of contamination. Could the authors clarify why CoNS positive cultures later considered as contaminations?
3. The authors need to detail the method/methods used for sample collection and bacteria isolation from CSF. The authors also need to provide information on how bacterial speciation was performed and verified.
4. In line 79-82, the authors described that the antimicrobial susceptibility testing (AST) was performed according to CLSI standard and a National Standards. Are both standards the same and what are the differences if not? More importantly, the presentation of the AST data in the manuscript should be improved. Figure 3 is not very informative since one can not easily tell the resistance rate in a specific CoNS species in a specific year. A table providing detailed percentages may be better.
5. It seems that the authors have access to the large amount of patient metadata, but not much analysis was carried out using these data in relation to the antimicrobial resistance in CoNS. It seems to be a missed opportunity.
6. The manuscript needs extensive revision for language and grammar.

Minor comments:

1. The "Trends of antimicrobial susceptibility tests" should change to "Trends of antimicrobial susceptibility", given that the work is investigating antimicrobial susceptibility, not the testing methods. (throughout the manuscript).
2. Line 83: "the breakpoints" should be "the range"
3. Line 97: Why only the first positive culture was analysed in an episode? Are the isolates from in subsequent positive culture the same with the first?
4. Line 107, the authors state that "after exclusion, 534 episodes from 519 patients were analysed"; in Line 111, the authors state 526 (98.5%) survived. This is confusing since the number of patients do not match.
5. A more detailed and clear legend is needed for each figure.
6. Table 3 needs to be improved. a) The table title is not proper. b) Presumably that the numbers in the year columns indicate the number of CoNS tested each year, could the author clarify in the note and add the resistant rate in each separate year?

Editor comment: I would have a native-English speaker proof read manuscript to improve readability.

Staff Comments:

Preparing Revision Guidelines

Please return the manuscript within 60 days; if you cannot complete the modification within this time period, please contact me. If you do not wish to modify the manuscript and prefer to submit it to another journal, please notify me of your decision immediately so that the manuscript may be formally withdrawn from consideration by Microbiology Spectrum.

In the study of M.M. Yi Ye et al. the retrospective analysis of dynamic antibiotic susceptibility of clinical isolates of coagulase-negative staphylococci (CoNS) isolated from the patients with neurological infections was performed. In order to differentiate the contaminated samples and true infections inclusion criteria were applied. Since the work is exclusively retrospective and the authors used the results of lab information systems only, no experimental studies were carried out. I've got the feeling that the manuscript is more like a report than a research study. Perhaps the work should be supplemented with some clinical data - a detailed description of clinical outcomes, its associations with different species of CoNS, as well as some additional clinical parameters for a proper comparison. Which treatment have the patients received? Was there any correlation with the antibiotic susceptibility data? What was the time period during the treatment after which the cerebrospinal fluid was free of CoNS?

Minor comments:

English should be revised.

Which method was used for CoNS identification?

Versions of CLSI, EUCAST should be specified.

Response to the reviewers' comments:

Reviewer #1:

Comments: In the study of M.M. Yi Ye et al. the retrospective analysis of dynamic antibiotic susceptibility of clinical isolates of coagulase-negative staphylococci (CoNS) isolated from the patients with neurological infections was performed. In order to differentiate the contaminated samples and true infections inclusion criteria were applied. Since the work is exclusively retrospective and the authors used the results of lab information systems only, no experimental studies were carried out. I've got the feeling that the manuscript is more like a report than a research study. Perhaps the work should be supplemented with some clinical data - a detailed description of clinical outcomes, its associations with different species of CoNS, as well as some additional clinical parameters for a proper comparison. Which treatment have the patients received? Was there any correlation with the antibiotic susceptibility data? What was the time period during the treatment after which the cerebrospinal fluid was free of CoNS?

Response: *We are grateful for the constructive suggestion that the clinical data should be included. If the work did not include clinical data, we could only assume that vancomycin and linezolid are safe in clinical practice. According to your suggestion, we described the treatment and clinical outcome, which helped prove the assumption. In conclusion, vancomycin and linezolid are safe in the laboratory and clinical practice.*

First, the treatment the patients received was described. Vancomycin (or

norvancomycin) was the commonest agent in clinical practice, followed by linezolid. The agents used in treatment were in correspondence with the results in the laboratory.

Second, we added a detailed description of the clinical outcome. Most patients had acceptable outcomes. We found that vancomycin was valuable in clinical practice. Moreover, the association of clinical outcome and bacteria species was discussed.

However, we did not analyze the time period during the treatment after which the cerebrospinal fluid was free of CoNS. There was only one positive culture for most episodes, and maybe the clinical outcome is a suitable study outcome in the analysis. The time period is valuable since free of bacteria in cerebrospinal fluid is an indicator of cure. We appreciate that you supplied a possible option about the study outcome.

1. Minor comment: English should be revised.

Response: *This is necessary for the international readership, and the English have been revised.*

2. Minor comment: Which method was used for CoNS identification?

Response: *All cerebrospinal fluid specimens were incubated until flagged as positive or for five days in BACTEC 9240 (Becton Dickinson, America) for 2012 Jan-2018 Sep or BacT/ALERT 3D (bioMérieux, France) for 2018 Oct-2020 Dec. The broth was analyzed automatically every 10 minutes bacteria from positive bottles, was Gram-stained, and sub-cultured onto solid medium using standard protocols.*

3. Minor comment: Versions of CLSI, EUCAST should be specified.

Response: *In this manuscript, the cerebrospinal fluid cultures and susceptibility tests were performed according to the newest versions of CLSI in the corresponding times, and the interpretive categories were performed according to M100, 31st ed.*

Reviewer #2:

Comment: This work by Ye et al. aims to investigate the trends of antimicrobial susceptibility in clinically significant coagulase-negative staphylococci (CoNS) isolated from cerebrospinal fluid (CSF) cultures in a Chinese hospital over a period of 9 years. The authors have concluded that no significant trend of antimicrobial susceptibility was observed in CoNS against the antimicrobials analysed in the study over the period of 9 years. However, the authors have also noted some visible changes, such as the increasing of vancomycin resistance and the 'disappearing' of lizezolid resistance. Although the study informs the overall situation of antimicrobial resistance in clinical CoNS in the Chinese hospital, much is needed to improve the methodology and data presentation in this manuscript in order to support these conclusions.

Response: *To be first, we really appreciate what you have done for our manuscript. The professional comments and kind advice impress us a lot. Your careful review helps us clarify the manuscript. Significantly, your major comment 5 about the clinical data reminds us of the addition of treatment and clinical outcome. After describing treatment and clinical outcome, we found that vancomycin and linezolid*

are safe in the laboratory and clinical practice.

1. **Major comment:** Most of the recruited 2342 patients were excluded (leaving one 519 patients). What is the rationale behind the patient exclusion criteria? For instance, if the authors were to investigate the antimicrobial susceptibility in clinically significant CoNS, why cultures from patients under 18 were excluded?

Response: *The reasons for the exclusion criteria were a. The aim was to discuss the CoNS causing healthcare-associated ventriculitis and meningitis in adults, and we should emphasize it by changing the title; b. According to a previous study, the main stage of healthcare-associated ventriculitis and meningitis was the first month after neurosurgeries(1). Therefore, the patients who accepted neurosurgeries in other hospitals in one month were excluded. c. The ventriculitis or meningitis is not acquired in the hospital for patients in the acute stage of open craniocerebral injury and pre-hospital intracranial infection. d. The patients without surgeries were excluded since we recruited neurosurgical patients. e. We cannot judge the infection and contamination in patients with incomplete data.*

2. **Major comment:** What is a contaminated episode? Figure 1 shows 1344 cultures were positive with CoNS amongst 1116 episodes, if I understand correctly. Then it shows 582 episodes were excluded because of contamination. Could the authors clarify why CoNS positive cultures later considered as contaminations?

Response: *The contaminated episode was defined as not infected episode and suited with one of the following criteria. I. The patient had no clinical features(2,*

3). II. The patient had clinical features, but bacteria were not pathogen(4), including IIa. The improvement occurred without sensitive antimicrobial agents(5); IIb. The duration of the sensitive antimicrobial agent was less than 72 hours out of medical reasons, and no re-ignition occurred(6); IIc. The sensitive antimicrobial agent was ineffective in the patient without special situations, including poor wound healing, implants, CSF leak, abscess, and ventriculitis, and the course was self-limiting without adjusting the antimicrobial agent(7). III. The patient had clinical features, and bacteria other than bacteria isolated were the pathogen. In this situation, the sensitive antimicrobial agent was ineffective in the patient without special situations, and the improvement occurred after adjusting the antimicrobial agent. There are two reasons for the diagnostic criteria. First, aseptic meningitis is common in neurosurgical patients, the clinical features are similar to bacterial meningitis(4), and the improvement is not related to antimicrobial agents(5, 7). The patients with clinical features did not accept sensitive antimicrobial agents because, for neurosurgical patients, the slightly changed body temperature and laboratory indicators in cerebrospinal fluid are acceptable when the patients are without other clinical manifestations(5). Second, the bacteria isolated may not be the pathogens in patients with healthcare-associated ventriculitis and meningitis because of the high contamination rate in cerebrospinal fluid cultures and low etiologic diagnosis rate in healthcare-associated ventriculitis and meningitis(8).

3. Major comment: The authors need to detail the method/methods used for

sample collection and bacteria isolation from CSF. The authors also need to provide information on how bacterial speciation was performed and verified.

Response: *CSF was collected from all patients with suspected healthcare-associated ventriculitis and meningitis. At the bedside, CSF was obtained by well-trained neurosurgical doctors using 70% isopropyl alcohol followed by 10% povidone-iodine as antiseptics. The commonest sampling method was lumbar puncture, followed by ventricular drainage and lumbar cistern drainage. All CSF specimens were immediately transferred to the clinical laboratory and were incubated until flagged as positive or for five days in BACTEC 9240 (Becton Dickinson, America) for 2012 Jan-2018 Sep or BacT/ALERT 3D (bioMérieux, France) for 2018 Oct-2020 Dec. The broth was analyzed automatically every 10 minutes bacteria from positive bottles, was Gram-stained, and sub-cultured onto solid medium using standard protocols.*

- 4. Major comment: In line 79-82, the authors described that the antimicrobial susceptibility testing (AST) was performed according to CLSI standard and a National Standards. Are both standards the same and what are the differences if not? More importantly, the presentation of the AST data in the manuscript should be improved. Figure 3 is not very informative since one can not easily tell the resistance rate in a specific CoNS species in a specific year. A table providing detailed percentages may be better.**

Response: *The national committee on clinical laboratory standards is the predecessor of the Clinical and Laboratory Standards Institute. In our institute,*

the standards changed according to the newest versions in the corresponding times. This means the unchanged part was performed according to the older versions. To avoid misunderstanding, we have changed the expression. Moreover, in table 3, we added data to describe the resistance rate every year.

- 5. Major comment: It seems that the authors have access to the large amount of patient metadata, but not much analysis was carried out using these data in relation to the antimicrobial resistance in CoNS. It seems to be a missed opportunity.**

Response: *It is a constructive suggestion that the work should be supplemented with clinical data. We described the treatment and clinical outcome. Moreover, the association of clinical outcome and bacteria species was discussed. We found vancomycin (or norvancomycin) was used to treat most CoNS-related healthcare-associated ventriculitis and meningitis. Most patients had acceptable outcomes. Therefore, vancomycin is safe in the laboratory and clinical practice.*

- 6. Major comment: The manuscript needs extensive revision for language and grammar.**

Response: *This is necessary for the international readership, and the English have been revised.*

- 1. Minor comments: The "Trends of antimicrobial susceptibility tests" should change to "Trends of antimicrobial susceptibility", given that the work is investigating antimicrobial susceptibility, not the testing methods. (throughout the manuscript).**

Response: *Combining with major comment 1, we decided to change the title to "Trends of antimicrobial susceptibility in clinically significant coagulase-negative staphylococci isolated from cerebrospinal fluid cultures in neurosurgical adults: a nine-year analysis." Moreover, the phrases were changed in the corresponding locations.*

2. Minor comments: Line 83: "the breakpoints" should be "the range"

Response: *We have changed it according to your comment.*

3. Minor comments: Line 97: Why only the first positive culture was analysed in an episode? Are the isolates from in subsequent positive culture the same with the first?

Response: Repeat isolates are usual and should be considered carefully. According to the CLSI M100, 31st ed (Page 43 VI. Development of Resistance and Testing of Repeat Isolates), development of resistance can occur within as little as three to four days and has been noted most frequently in Enterobacter (including Klebsiella aerogenes), Citrobacter, and Serratia spp. with third-generation cephalosporins, in Pseudomonas aeruginosa with all antimicrobial agents, and in staphylococci with fluoroquinolones. For Staphylococcus aureus, vancomycin-susceptible isolates may become vancomycin intermediate during the course of prolonged therapy. In the manuscript, only vancomycin, norvancomycin, linezolid, and teicoplanin were used. Therefore, we only discussed the first isolate.

4. Minor comments: Line 107, the authors state that "after exclusion, 534 episodes from 519 patients were analysed"; in Line 111, the authors state 526

(98.5%) survived. This is confusing since the number of patients do not match.

Response: *The number should be expressed clearly, meaning the 526 (98.5%) episodes rather than 526 patients survived. It is worth noting that we have changed the definition of clinical outcomes in this study. In the last manuscript, the clinical outcome was judged after all hospitalizations in our hospital. Differently, in this manuscript, the clinical outcome was judged after the hospitalization in which the patient had healthcare-associated ventriculitis and meningitis because the treatment was described.*

5. Minor comments: A more detailed and clear legend is needed for each figure.

Response: *This is necessary for the readership, and we have tried to give more detailed and clear legends.*

6. Minor comments: Table 3 needs to be improved. a) The table title is not proper. b) Presumably that the numbers in the year columns indicate the number of CoNS tested each year, could the author clarify in the note and add the resistant rate in each separate year?

Response: *It is a valuable suggestion to clarify the number, and we have changed it according to your comment.*

References

1. Shi ZH, Xu M, Wang YZ, et al. Post-craniotomy intracranial infection in patients with brain tumors: a retrospective analysis of 5723 consecutive patients. *Br J Neurosurg.* 31(1). England,2017. 5-9.

2. Boysen MM, Henderson JL, Rudkin SE, et al. Positive cerebrospinal fluid cultures after normal cell counts are contaminants. *J Emerg Med.* 37(3). United States,2009. 251-6.
3. Steinbok P, Cochrane DD, Kestle JR. The significance of bacteriologically positive ventriculoperitoneal shunt components in the absence of other signs of shunt infection. *J Neurosurg.* 84(4). United States,1996. 617-23.
4. Mount HR, Boyle SD. Aseptic and Bacterial Meningitis: Evaluation, Treatment, and Prevention. *Am Fam Physician.* 96(5). United States,2017. 314-322.
5. Forgacs P, Geyer CA, Freidberg SR. Characterization of chemical meningitis after neurological surgery. *Clin Infect Dis.* 32(2). United States,2001. 179-85.
6. Zarrouk V, Vassor I, Bert F, et al. Evaluation of the management of postoperative aseptic meningitis. *Clin Infect Dis.* 44(12). United States,2007. 1555-9.
7. Bihan K, Weiss N, Théophile H, et al. Drug-induced aseptic meningitis: 329 cases from the French pharmacovigilance database analysis. *Br J Clin Pharmacol.* 85(11) ,2019. 2540-2546.
8. Srihawan C, Castelblanco RL, Salazar L, et al. Clinical Characteristics and Predictors of Adverse Outcome in Adult and Pediatric Patients With Healthcare-Associated Ventriculitis and Meningitis. *Open Forum Infect Dis.* 3(2) ,2016. ofw077.

December 15, 2021

Prof. Guangzhi Shi
Beijing Tiantan Hospital, Capital Medical University
No.119, South Fourth Ring West Road, Fengtai District, Beijing
Beijing
China

Re: Spectrum01462-21R1 (Trends of antimicrobial susceptibility tests in clinically significant coagulase-negative staphylococci isolated from cerebrospinal fluid cultures: a nine-year analysis)

Dear Prof. Guangzhi Shi:

Link Not Available

Sincerely,

Tomefa Asempa

Journals Department
Editor comments:

1. Several sentences not clear. For example line 70 "It is not suitable to represent clinically significant CoNS with CoNS isolated." Line 43 "Vancomycin and linezolid will be safe in the CoNS-related HCAVM" Safe in treatment? please be specific. I understand language challenges but strongly encourage you to utilize an English Editing service review. This is critical for meeting the standards of publication in ASM. A few can be found here <https://journals.asm.org/language-editing-services>

2. Please revise aim in abstract. Aim was not to describe the tests.

3. Line 99. range of MICs are specific to an antibiotic and are expected to vary. please clarify the range you report? what abx are those for? Can take sentence out.

4. Line 229 and 196. What does it mean for an abx to be safe in the laboratory?

5. Figure 3. where are the P values? what is Agent: P-value?

6. What constitutes "other" in Table 4? include the number of episodes in Table 4.

7. you provide several resistance rates in the results section of abstract? what year or time are you referring to?

This is the final review and will not be reviewed again if modifications inadequate.

Staff Comments:

Preparing Revision Guidelines

Please return the manuscript within 60 days; if you cannot complete the modification within this time period, please contact me. If you do not wish to modify the manuscript and prefer to submit it to another journal, please notify me of your decision immediately so that the manuscript may be formally withdrawn from consideration by Microbiology Spectrum.

Response to the reviewers' comments:

Editor comments:

1. Several sentences not clear. For example line 70 "It is not suitable to represent clinically significant CoNS with CoNS isolated." Line 43 "Vancomycin and linezolid will be safe in the CoNS-related HCAVM" Safe in treatment? please be specific.

I understand language challenges but strongly encourage you to utilize an English Editing service review. This is critical for meeting the standards of publication in ASM. A few can be found here <https://journals.asm.org/language-editing-services>

Response: *Thanks for your help. We have chosen AJE, one of your recommendations, and we feel your comment is valuable for this work.*

2. Please revise aim in abstract. Aim was not to describe the tests.

Response: *We have changed it according to your comment.*

3. Line 99. range of MICs are specific to an antibiotic and are expected to vary. please clarify the range you report? what abx are those for? Can take sentence out.

Response: *The MICs were for all antibiotics analyzed, and we have decided to take the sentence out. Thanks for the advice.*

4. Line 229 and 196. What does it mean for an abx to be safe in the laboratory?

Response: *An abx safe in the laboratory means that the abx is not resistant according to the antimicrobial susceptibility tests. And we have changed the expression.*

5. Figure 3. where are the P values? what is Agent: P-value?

Response: *Thanks for the comment. We have added the P-values in Figure 3, and the P-values are for the trends of resistance rates, which help prove "There were no*

significant trends in antimicrobial susceptibility for any agents analyzed during the nine-year period".

6. What constitutes "other" in Table 4? include the number of episodes in Table 4.

Response: *We have changed it according to your comment.*

7. you provide several resistance rates in the results section of abstract? what year or time are you referring to?

Response: *The resistance rates were for the nine years, and we have clarified the expression by adding "over the nine years."*

January 4, 2022

Prof. Guangzhi Shi
Beijing Tiantan Hospital, Capital Medical University
No.119, South Fourth Ring West Road, Fengtai District, Beijing
Beijing
China

Re: Spectrum01462-21R2 (Trends of antimicrobial susceptibility tests in clinically significant coagulase-negative staphylococci isolated from cerebrospinal fluid cultures: a nine-year analysis)

Dear Prof. Guangzhi Shi:

Thank you for utilizing editing service. Manuscript reads well.

Your manuscript has been accepted, and I am forwarding it to the ASM Journals Department for publication. You will be notified when your proofs are ready to be viewed.

Sincerely,

Tomefa Asempa
Editor, Microbiology Spectrum
